# The Post-Curing of Waterborne Polyurethane–Acrylate Composite Latex with the Dynamic Disulfide-Bearing Crosslinking Agent

**DOI:** 10.3390/molecules28248122

**Published:** 2023-12-15

**Authors:** Haotian Zhang, Rihui Liang, Qianshu Wang, Wenbo Luan, Jun Ye, Teng Qiu, Xinlin Tuo

**Affiliations:** 1Key Laboratory of Carbon Fiber and Functional Polymers, Ministry of Education, Beijing University of Chemical Technology, Beijing 100029, China; zhanghaotian108@163.com (H.Z.); 15913169262@163.com (R.L.); 2022200375@buct.edu.cn (Q.W.); 2023200385@buct.edu.cn (W.L.); 2Beijing Engineering Research Center of Synthesis and Application of Waterborne Polymer, Beijing University of Chemical Technology, Beijing 100029, China; 3Key Laboratory of Advanced Materials (MOE), Department of Chemical Engineering, Tsinghua University, Beijing 100084, China; tuoxl@mail.tsinghua.edu.cn

**Keywords:** waterborne, polyurethane–acrylate, latex, self-healing, dynamic network

## Abstract

The development of a dynamic network for commodity polymer systems via feasible methods has been explored in the context of a society-wide focus on the environment and sustainability. Herein, we demonstrate an adaptive post-curing method used to build a self-healable network of waterborne polyurethane–acrylate (WPUA) composite latex. The composite latex was synthesized via the miniemulsion polymerization of acrylates in the dispersion of waterborne polyurethane (PU), with commercial acetoacetoxyethyl methacrylate (AAEM) serving as the functional monomer. Then, a dynamic disulfide (S–S)-bearing diamine was applied as the crosslinking agent for the post-curing of the hybrid latex via keto-amine condensation, which occurred during the evaporation of water for film formation. It was revealed that the microphase separation in the hybrid films was suppressed by the post-curing network. The mechanical performance exhibited a high reliability as regards the contents of the crosslinking agents. The reversible exchange of S–S bonds meant that the film displayed associative covalent-adaptive networks in the range of medium temperature in stress relaxation tests, and ≥95% recovery in both the stress and the strain was achieved after the cut-off films were self-healed at 70 °C for 2 h. The rebuilding of the network was also illustrated by the >80% recovery in the elongation at break of the films after three crushing–hot pressing cycles. These findings offer valuable insights, not only endowing the traditional WPUA with self-healing and reprocessing properties, but broadening the field of study of dynamic networks to polymer hybrid latex.

## 1. Introduction

The discovery and commercialization of polymer materials have provided great opportunities in the contexts of both industry and human lives. However, the progress has also led to increasing pressures on the environment and resources. Dynamic covalent polymer networks (DCPNs) are highlighted as one of the more promising solutions. The construction of topologically stable and chemically reversible polymer networks combines the advantages of thermosets and thermoplastics [1,2]. Various types of dynamic covalent bonds (DCBs) have been studied for this purpose, including hindered urea [3,4], boronic ester [5,6,7], ester [8,9], imine [10,11], Diels–Alder [12,13], silyl ether [14,15], disulfide [16,17,18,19] and olefin metathesis [20,21], etc. The dissociative or associative mechanisms and the underlying physics at different timescales have also been progressively revealed [22]. However, challenges still exist in relation to the complexity of the polymer aggregation structures.

Polyurethane (PU) is one of the most widely used commodity polymers. The customizable chain structures make PU an attractive matrix for DCPN studies. Such DCPNs are recognized as special since the network tends to behave as associative even when the dissociative type of DCB is used. The reason is related to the microphase separation of the hard/soft segments and the intrinsic H-bonding networks [23]. Besides PU, polyacrylate (PAC) is another type of general resin. The DCPNs of PAC are prepared in the literature through solution polymerization/reactive extrusion [24], reversible addition–fragmentation chain transfer (RAFT) polymerization [25], and bulk polymerization or UV-curing [26,27]. Combining the advantages of both PU and PAC, such as toughness, elasticity, resistance to different conditions, and cost-effectiveness, the synthesis and application of polyurethane–acrylate is an attractive issue nowadays [28,29,30]. Corresponding DCPNs of polyurethane–acrylate are also seen in the literature, but their study is still limited, and most of the composites are prepared under homogeneous solution polymerization [31] or UV curing with the active diluents [32].

In contrast to the polymerization carried out under homogeneous conditions, there is a special interest in the heterophase polymerization (HPP) represented by emulsion polymerization. The latex composites prepared by HPP are much welcome in the fields of coatings and adhesives. Moreover, the use of water as the reaction medium is environmentally friendly and results in “green products”. By introducing PU moiety into the HPP of PAC via blending or grafting, waterborne polyurethane–acrylate (WPUA) composite latex can be synthesized [33,34]. However, little is known about the effects when the DCBs are introduced into the composite latex of WPUA.

Among various DCBs, the metathesis of aromatic disulfide (S–S) bonds under different kinds of stimuli has been extensively investigated [35,36,37]. In our previous work [38], we have demonstrated one post-curing strategy for the DCPN building of the PAC latex. In this work, we put forward the strategy as a feasible method to build the DCPN of WPUA composite latex, which was synthesized via miniemulsion polymerization with the functional monomer of methacrylate acetoacetoxyethyl (AAEM). The network was formed by the post–curing of the latex film through keto-amine condensation with 4,4′-dithiodianiline (DTDA) serving as the diamine, and the dynamic S–S was therefore linked in simultaneously. Condensation would not occur in the aqueous medium, which permitted the mixture of DTDA and as-synthesized latex to remain stable in the one-pot two-component form. However, the microphase structures, the mechanical properties, and especially the dynamic responses to the elevated temperatures of the consequent composite films were dramatically changed by the disulfide-bearing network, which issues are discussed in detail in the following sections.

## 2. Results and Discussion

### 2.1. Synthesis of WPUA–AAEM Latex

The synthesis of AAEM-decorated composite latex, denoted as WPUA–AAEM *x*%, is illustrated in Figure 1a. The amphiphilic WPU is situated at the interface of the miniemulsion droplets, similar to the Pickering particles, to stabilize the droplets synergizing with the emulsifiers. The polymerization in the miniemulsion proceeded stably in the *x*% range from 0 to 20%. The characterization results for the composite latex are summarized in Table 1. It can be seen that acrylic monomer conversions greater than 95% were achieved in all the syntheses. The *Z*–average particle sizes increased for the samples with enhanced *x*% from 0 to 20%, but were well controlled below 150 nm. Even for the sample with *x*% = 20%, the PDI was similar to that of *x*% = 0. All latex samples exhibited a milky white appearance, and good centrifugation stabilities and storage stabilities were achieved.

A comparison of the stabilities of WPUA–AAEM 0%, WPUA–AAEM 20%, and WPUA–AAEM–DTDA 20% is displayed in Figure 2. In Figure 2a–c, all the *BS*% within the height range from 10 mm to 40 mm remained at about 16%, indicating that the particle sizes were consistent. It can be seen that the curves showed only slight differences during 12 h of storage at 60 °C for all the samples, suggesting that there were no great changes in the size and distribution of the particles dispersed in the latex. The decoration of the latex with AAEM, as well as the following addition of the crosslinking agent DTDA, did not disrupt the stability of the composite latex. In Figure 2d, the TEM image of WPUA–AAEM 15% latex particles is presented. The morphology of these particles displayed a typical core–shell structure. The grafting of PU to PAC is illustrated in the Appendix A, where the end-capped C=C was consumed during the polymerization of the PAC phase.

### 2.2. The Film Formation and Curing

The stable WPUA–AAEM–DTDA latex was cast into films. The spontaneously formed post-curing dynamic composite network is sketched in Figure 1b. The keto–amine condensation and the incorporation of the aromatic S–S in the film has been revealed by FTIR. The spectra of WPUA–AAEM 0% and WPUA–AAEM–DTDA 20% are shown in Appendix A from 4000 cm^−1^ to 400 cm^−1^. The typical bands for PU and PAC, such as those at 3210 cm^−1^ to 3430 cm^−1^ assigned to the N–H stretching vibrations, 2940 cm^−1^ and 2854 cm^−1^ to the C–H stretching vibrations, 1800 cm^−1^ to 1620 cm^−1^ to the C=O stretching vibrations, 1531 cm^−1^ to the N–H bending vibrations, 1465 cm^−1^ to the –CH_2_– bending vibrations, 1365 cm^−1^ and 1396 cm^−1^ to the non-symmetrical and symmetrical –CH_3_ bending vibrations, 1240 cm^−1^ to the C–O stretching vibrations, and 1112 cm^−1^ to the C–O–C stretching vibrations, are all observable. In comparison with WPUA–AAEM 0%, a distinctive band at 517 cm^−1^ was detected in the zoom-in spectrum of WPUA–AAEM–DTDA 20% in Figure 3a. The band is characteristic of S–S. Its emergence reveals the linkage of aromatic S–S bonds into the polymer network through crosslinking reactions, as expected. Furthermore, the band at 1652 cm^−1^ in Figure 3b broadened due to the formation of unsaturated double bonds (–CH=CH–NH–), proving evidence of the keto–amine condensation between AAEM and DTDA. In addition, the bands for benzene rings and C–S bonds attached in DTDA can be observed in Figure 3b at 1592 cm^−1^ and 1493 cm^−1^, respectively.

The UV-vis spectra of WPUA–AAEM–DTDA *x*% films are shown in Figure 3c. It can be observed that the WPUA–AAEM 0% film showed the transmittance of ~85% in the range of 400 nm to 800 nm. As *x*% increased, the transmittance gradually decreased in an almost parallel manner, likely due to the increased scattering as a consequence of the enlarged particle sizes. The transmittance dropped to zero at ~400 nm, possibly because of the incorporation of DTDA with chromophores. The yellow color of DTDA in solution and the darkening color in the films with the increase in DTDA dosage can be seen in Appendix A. The photograph of WPUA–AAEM 10% film is shown as the inset in Figure 3c. The composite film presented as uniform, continuous and transparent without macroscopic phase separation. The TGA profiles of WPUA–AAEM 0% and 20% are provided in Figure 3d. The introduction of the AAEM–DTDA network did not show an obvious impact on the thermal stability of the composite films.

The swelling tests of WPUA–AAEM–DTDA *x*% films were conducted in THF for 24 h at room temperature, and the results are presented in Figure 4a. The apparent gel content in THF (*gel*%) was ~73% for WPUA–AAEM 0%, which would be related to the intrinsic physical and chemical crosslinking points contributed by H-bonding interactions and the grafting reactions [33], respectively. As *x*% increased from 0 to 5%, the *gel*% increased to a maximum of over 90%, confirming the post-curing reaction of AAEM–DTDA. Correspondingly, the apparent swelling degree in THF in 24 h (*swelling*%) also decreased to the minimum of 250%. But the *gel*% decreased again with further increases in *x*%. The *gel*% of WPUA–AAEM–DTDA 20% was even slightly lower than that of WPUA–AAEM 0%. The corresponding *swelling*% also increased to about 600% as *x*% increased from 5% to 20%. It has been suggested that the dynamic exchange of substances would occur in the associative dynamic network based on aromatic S–S bonds in organic solvents [39]. Moreover, the consumption of the keto groups of AAEM by side-reactions could not be ignored. The diamine would be present in excess, and promotes the dynamic exchange reactions. In this respect, the low content of DTDA would mainly contribute to the formation of network points, while the dynamic effect would become more pronounced at higher contents. The comprehensive result is the decreased *gel*% and increased *swelling*% when *x*% was larger than 5% in the 24 h of swelling in THF. The photographs of WPUA–AAEM–DTDA *x*% films after 24 h of immersion in THF are provided in Appendix A. All the films sustained their shapes in the swollen state.

The dynamic network exchange is illustrated through stress relaxation (SR) in Figure 4b. The relaxation time (*τ*) was observed within 10 min and accelerated with increasing temperatures. Based on the Maxwell model for viscoelastic fluids, the *τ* was linearly fitted to the temperature based on the Arrhenius equation [40], and the results are shown in Figure 4c. The activation energy (*E*_a_) was calculated through fitting, resulting in 31 kJ/mol, which value is much lower than that of all-PAC latex networks crosslinked by DTDA, as prepared by Gong et al. [41]. The difference may stem from their low relative contents of AAEM/DTDA, which were limited to within 7%, accounting for the total weight of PAC. In this research, the dosage of the AAEM/DADA was not significantly increased in relation to the total resin weight (PU + PAC). But, since the post-curing was contributed mainly by the PAC phase, the possibility of locally increased DCB exchange would be the reason for the accelerated chain exchange reactions.

The microphase structure is illustrated by the tan*δ*–temperature spectra in Figure 4d, characterized by DMA. It can be observed that all the curves exhibited three transitions. The peak in the range of −10 °C to 0 °C is ascribed to the glass transition temperature (*T*_g_) of the PAC phase, denoted as *T*_g, PAC_. The peak near 60 °C is assigned to the *T*_g_ of the hard segmental phase of PU (*T*_g, PUH_). The small sub-peak near −40 °C is assigned to the *T*_g_ of the soft segmental phase of PU (*T*_g, PUS_). *T*_g, PUS_ remained almost unchanged with the increase in *x*%, indicating that the cross-linking had little effect on the soft segmental phases. The position of *T*_g, PUH_ stood also at almost the same place, but the peak broadened with the increase in *x*%. At the same time, the *T*_g, PAC_ gradually shifted toward *T*_g, PUH_. Both the shortened distance between the two peaks and the elevation of the valley between them suggest suppression on the microphase separation of PAC and the hard segmental domains of PU. In this way, the composite film exhibited was more and more similar to the latex interpenetration network (LIPN), and the damping temperature range with tan *δ >* 0.3 [42] spanned from −8 °C to above 100 °C for the sample of WPUA–AAEM–DTDA 20%. The corresponding storage modulus spectra are provided in Appendix A.

To further illustrate the contribution of post-curing, two other samples were prepared with DTDA fixed only in the PU phase (WPU–2%DTDA–PAC) through copolymerization, and in both PAC and PU phases (WPU–2%DTDA–PAC–DTDA 20%) through the post-curing of WPU–2%DTDA–PAC by DTDA, respectively. The comparison results are shown in Figure 4e. It can be seen that the *T*_g, PUH_ peak moved to the highest temperature, and the most distinctive three peaks were observed on the sample of WPU–2%DTDA–PAC without post-curing. In comparison with WPU–2%DTDA–PAC, the formation of the post-curing network and its forced compatibilization effects on the three phases in WPUA–AAEM–DTDA 20% are indicated by the shifts of the peaks, with *T*_g, PUH_ and *T*_g, PAC_ to lower temperatures, and *T*_g, PUS_ to higher temperatures. However, the most extensive suppression of the phase-separation was seen in the sample of WPU–2%DTDA–PAC–DTDA. The increased component similarity as well as the chain exchange through the S–S metathesis resulted in the merger of the peaks of *T*_g, PUH_ and *T*_g, PAC_ into a strong and broad one in the spectrum.

### 2.3. The Mechanical Properties, Self-Healing and Reprocessing

The mechanical properties and self-healing of WPUA–AAEM–DTDA *x*% films are depicted in Figure 5. The typical stress-strain curves are seen in Figure 5a. In comparison with WPUA–AAEM 0%, all the post-cured films showed an increased strength/modulus and decreased elongation at break. The quantitative variations are plotted in Figure 5b as a function of *x*%. The tensile strength of WPUA–AAEM 0% was 7.9 MPa, with an elongation at break of 874%. The enhanced strength (10.4 MPa) and the minimum elongation at break (359%) was reached at *x*% = 5 as the effect of the post-curing. However, as *x*% further increased, the elongation at break gradually increased. The tensile strength also increased, reached the maximum of 10.6 MPa at *x*% = 10%, and descended to ~ 10 MPa at *x*% = 15% and 20%. The enhancement in elongation at break would be related to the increased content of flexible S–S bonds. Moreover, their dynamic exchange would also contribute to the energy dissipation. It can be seen in Figure 5c that all the other samples showed an elevated fracture energy compared to that of *x*% = 5%, calculated through the integration on the underline area of the tensile curves. For the sample with *x*% = 20%, the modulus was almost doubled in comparison with that of *x*% = 0%, but with no sacrifice in the toughness indicated by the quite close fracture energies of the two samples. In this way, the post-curing of AAEM–DTDA on the composite WPUA latex tends to strike a balance between the improvements in strength and toughness.

All the samples were able to recover from the cut-off damage at 70 °C, except AAEM–DTDA (*x*% = 0). The process is illustrated in Appendix A. It can be seen in Figure 6a that the mechanical performance of the self-healed sample with *x*% = 5 was quite poor despite having almost the best original strength. The shapes of the stress–strain curves for the self-healed samples with *x*% = 10% were close to the original ones, and the strength as well as the strain increased with the elongation of the self-healing time, following the trace of the original stress–strain curve in Figure 6b. Elevation in the curve slope was seen for the samples with *x*% = 15% (Figure 6c) and 20% (Figure 6d), suggesting the reconfiguration of the composite network structures. The best results were achieved at the highest AAEM/DTDA content with *x*% = 20%, where a high elongation at break was observed closest to the original one, indicating the reconstruction of the dynamic network after two hours of self-healing. The stress–strain curves of the original and self-healed samples of WPU–2%DTDA–PAC are provided in Appendix A. It can be seen that the original strength of the films was enhanced to 15.3 MPa with the elongation at break of 976%. However, the consequently self-healed sample presented a strength of only 5.8 MPa with an elongation of 425%, much poorer than that of the self-healed WPUA–AAEM–DTDA 20%. WPU–2%DTDA–PAC–DTDA 20% also showed poor mechanical performance after self-healing in Appendix A. However, improvements in the mechanical performance were not observed like that in the WPU–2%DTDA–PAC.

The self-healing efficiencies of WPUA–AAEM–DTDA *x*% films on the tensile strength (*H_σ_*) and the elongation at break (*H_ε_*) after 2 h of healing at 70 °C are plotted as a function of *x*% in Figure 6e. The efficiencies increased simultaneously with the increasing *x*%. It can be observed that both the *H_σ_* and *H_ε_* at *x*% = 5% were less than 20%. The increasing of *H_σ_* and *H_ε_* to ~80% for *x* = 10% and 15% clearly suggests that the increased contents of dynamic S–S bonds facilitate the reconstruction of the WPUA post-curing network from the damage. At *x*% = 20%, the *H_σ_* of the corresponding film exceeded 100%, and *H_ε_* reached 95%, demonstrating an excellent self-healing ability.

The stress–strain curves of the WPUA–AAEM–DTDA 20% films after different reprocessing cycles are shown in Figure 7a. Although not as much as that of the original film, the reprocessed samples all maintained 600% of elongation at break, suggesting that the dynamic exchange of the reversible interaction was adaptable enough to largely sustain their networks. The degeneration in the film’s strength would possibly be related to the changes in microphase separation in competition with the ordered, disordered and hybrid H-bonding interactions in the composite film during the reprocessing. The failure of the permanent cross-linking points between the PU and PAC could also possibly take some responsibility for the decreased performance. The *H_σ_* and *H_ε_* after different crushing–reprocessing circles are plotted in Figure 7b. It can be observed that the *H_ε_* of the WPUA–AAEM–DTDA 20% film remained at 80% even after three reprocessing cycles.

## 3. Materials and Methods

### 3.1. Materials 

Isophorone diisocyanate (IPDI) was purchased from Bayer, GER. Polytetramethylene ether glycol (PTMG, 2000 g/mol) was purchased from Jiangsu Jiaren Chemical Works Co., Ltd., Beijing, China. AAEM, DTDA and *N*-Methyl-2-pyrrolidone (NMP) were purchased from Shanghai Aladdin Biochemical Technology Co., Ltd., Beijing, China. 2-hydroxyethyl methacrylate (HEMA) and 2,2-di(hydroxymethyl) butyric acid (DMBA) were purchased from Shanghai Macklin Biochemical Technology Co., Ltd., Beijing, China. 1,4-butanediol (BDO) was purchased from Tianjin Fuchen Chemical Reagent Co., Ltd., Beijing, China. Dibutyltin dilaurate (DBTDL) was purchased from Tianjin Guangfu Technology Development Co., Ltd., Beijing, China. Methyl methacrylate (MMA) and 2-ethylhexyl acrylate (EHA) were purchased from Qilu Petrochemical Co., Ltd., Beijing, China. Hexadecane (HD) was purchased from Hubei Xinrunde Chemical Co., Ltd., Beijing, China. Self-made deionized (DI) water was used in the synthesis. All other solvents and emulsifiers were commercial products purchased from the market.

### 3.2. Characterization

The particle size and distribution were tested by the dynamic laser scattering (DLS) of a Zetasizer NanoZS, Malvern, UK. Fourier transform Infrared (FTIR) spectra were recorded by a Tensor 37, Bruker, Billerica, MA, USA. Attenuated total reflection FTIR (ATR–FTIR) spectra were collected using a Nicolet IS5, Thermo Fisher, Waltham, MA, USA. The morphology of WPUA latex particles was observed by an HT-7700 (Hitachi, Tokyo, Japan) transmission electron microscope (TEM). The latex was stained by 3 wt. % of sodium phosphotungstate solution before characterization. The optical stability analysis was carried out on a Turbiscan Lab, Formulation, FR. The backscattering (*BS*%) of the latex to an 880 nm near-infrared light laser was recorded per 40 microns along the height of the sample tube, and the scan was repeated every 30 min. The test temperature was 60 °C. UV–visible (UV–vis) spectra were recorded on a UV–3150, Shimadzu, Kyoto, Japan. Dynamic mechanical analysis (DMA) was carried out on a Q800, TA, US. The film samples were preloaded with a force of 0.01 N before characterization. The spectra were recorded in the temperature range from −100 °C to 100 °C at the ramping rate of 3 °C/min and frequency of 1 Hz. The stress relaxation was characterized by this DMA at a strain of 3%. The tensile properties of the films were tested using a CMT4304 electronic universal testing machine in accordance with the method specified in GB/T 1040.3–2006 [43]. 

### 3.3. Synthesis of WPUA Composite Latex

Two steps were involved in the synthesis.

Step 1. The synthesis of methacrylate end-capped waterborne polyurethane (MWPU) dispersion.

In a typical synthesis, 11.11 g of IPDI, 20 g of PTMG and 2.44 g of DMBA dissolved in ~3 g of NMP were charged in a 500 mL four-neck round-bottom flask equipped with a mechanical stirrer, a thermometer, a reflux condenser, a nitrogen inlet and an outlet. The system was purged with N_2_ for 30 min before the addition of 0.2 wt. % of DBTDL. The reaction temperature was then raised to 80 °C. After 3 h, 1.36 g of BDO was added, and the reaction was continued for 1 h. A small amount of acetone was added for viscosity adjustment. The temperature was then reduced to 70 °C, and 2.34 g of HEMA was added to end-cap the residual NCO. After 2 h, the reaction temperature was lowered to 50 °C, and 1.67 g of TEA was added for neutralization in 30 min. An appropriate amount of water was added under accelerated stirring for 30 min of emulsification. The dispersion was treated under vacuum to remove the small amount of solvent. The final MWPU dispersion had a solid content of ~30%.

Step 2. The synthesis of WPUA composite latex.

In the synthesis, 5.5 g of MMA, 11 g of EHA, and proper amounts of HD and AAEM were added to the aqueous solution of the emulsifiers. Under constant stirring at 500 rpm, the mixture was thoroughly dispersed for 10 min. Subsequently, the prepared MWPU dispersion was added to the mixture, and the stirring continued for 10 min, resulting in the coarse emulsion. The miniemulsion was obtained after the coarse emulsion was sonicated for 5 min using a sonication tip at 50% power, with the operation manner of 30 s on and 15 s off. The resulting miniemulsion was then transferred to a 500 mL four-neck round-bottom flask equipped with a mechanical stirrer, a thermometer, a reflux condenser, a nitrogen inlet and an outlet. The temperature was raised to 80 °C, and the initiator of APS was added. Four hours later, the temperature was lowered to room temperature. The pH was adjusted to 8–9 using ammonia water. The product was named as WPUA–AAEM *x*%, where *x*% is designated as the mass fraction of AAEM in the acrylic monomers fed in the second step. The conversion of the acrylic monomers was determined by the gravimetric method.

### 3.4. Film Formation and Post-Curing

DTDA was dissolved in a small amount of ethanol and added to the latex of WPUA–AAEM *x*% in the molar ratio of AAEM:DTDA = 2:1. The mixture was cast into a PTFE mold after 15 min of magnetic stirring. The film was dried at room temperature for 3 h before it was transferred to an oven and further dried at 60 °C for 24 h. The films were named WPUA–AAEM–DTDA *x*%, where *x*% represents the mass fraction of AAEM in the acrylic monomers fed in the second step, as in WPUA–AAEM *x*%.

### 3.5. The Self-Healing and Reprocessing

At room temperature, dumbbell-shaped specimens prepared for tensile tests were cut off from the center using a surgical scalpel. The severed sections were reattached and fixed with small clamps. The temporarily attached specimens were placed in an oven at 70 °C for a designated period. Healed specimens were obtained after 12 h of cooling at room temperature. The healing efficiency (*H*) was evaluated based on tensile strength (*σ*) and fracture elongation (*ε*), as provided in Equations (1) and (2), respectively.
*H_σ_* = *σ*/*σ*_0_ × 100%(1)
*H_ε_* = *ε*/*ε*_0_ × 100%(2)
where *σ*_0_ and *ε*_0_ represent the initial tensile strength and fracture elongation of the original specimens, while *σ* and *ε* denote the tensile strength and fracture elongation of the repaired specimens, respectively.

For the reprocessing test, the film was fragmented at first. The pieces were collected and hot-pressed for 2 min at 120 °C, 5 MPa. The recovery efficiencies were also evaluated using Equations (1) and (2).

## 4. Conclusions

A series of WPUA–AAEM *x*% composite core–shell latexes were synthesized by miniemulsion polymerization with the incorporation of AAEM. The introduction of dynamic disulfide bonds into the composite network was achieved by adding DTDA to the dispersion for the post-curing of the latex film via the keto–amine condensation, as suggested by the FTIR, TGA and DMA characterization results. The formation of the aromatic S–S incorporated network did not show a strong impact on the thermal stability, but forced compatibilization was observed on the different microphases of PU and PAC. The associative type of dynamic network was indicated by the swelling–relaxation time behavior in the moderate temperature range (30 °C to 70 °C). At low dosages of the AAEM–DTDA, the post-curing effect was mainly seen on the improvement in the tensile strength and solvent resistance, similar to traditional curing systems. However, the dynamic exchange would dominate the behavior of the post-cured latex films as the *x*% increased to high levels. Besides the contents, the location of the dynamic S–S in the composite latex would also influence the recovery of the dynamic network. Under the optimum condition, with 20% of AAEM/DTDA incorporated mainly in the PAC phase, the film exhibited a tensile strength of 9.9 MPa and an elongation at break of 734%. It was able to self-heal from the cut-off damage at 70 °C for 2 h, showing a high self-healing efficiency greater than 95%, and more than 80% retention of the mechanical performance was observed through two cycles of crushing–reprocessing. 

## Figures and Tables

**Figure 1 molecules-28-08122-f001:**
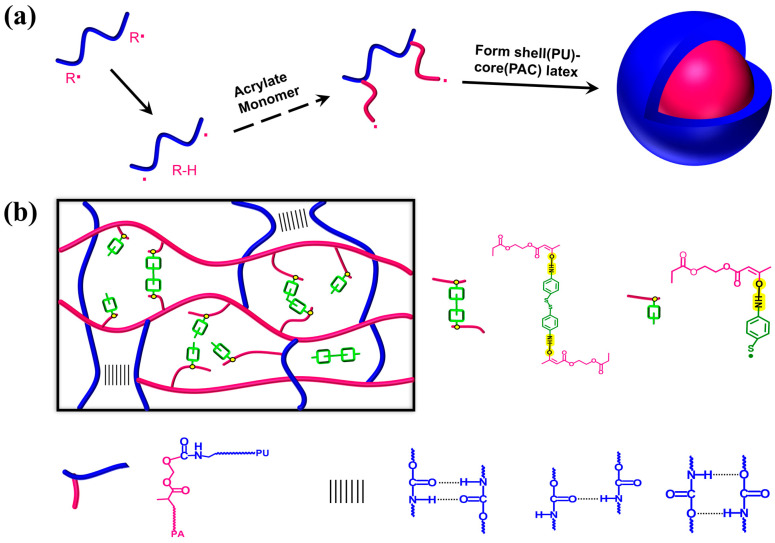
(**a**) The synthesis of WPUA latex. (**b**) Schematic diagram of WPUA post-curing networks.

**Figure 2 molecules-28-08122-f002:**
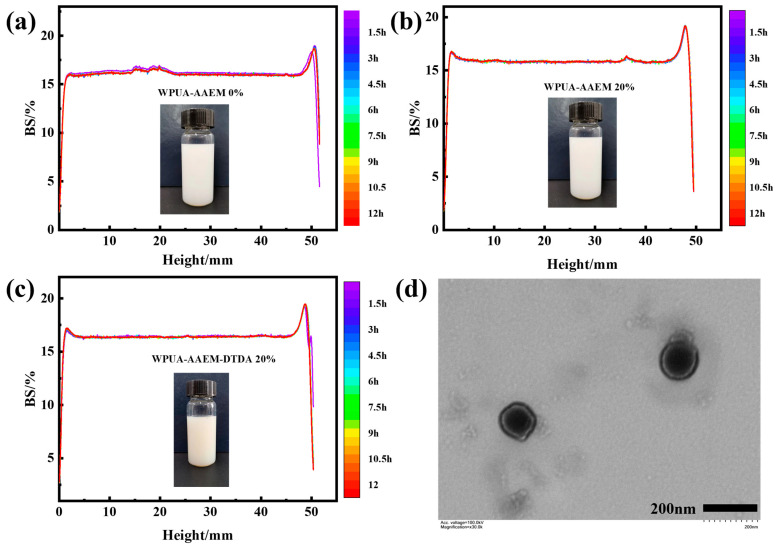
(**a**–**c**) The variation in *BS*% as a function of sample height for WPUA–AAEM 0%, WPUA–AAEM 20%, and WPUA–AAEM–DTDA 20%, respectively. Different colored curves represent samples subjected to varying storage durations at 60 °C. (**d**) The TEM image of the WPUA–AAEM 15% latex particles.

**Figure 3 molecules-28-08122-f003:**
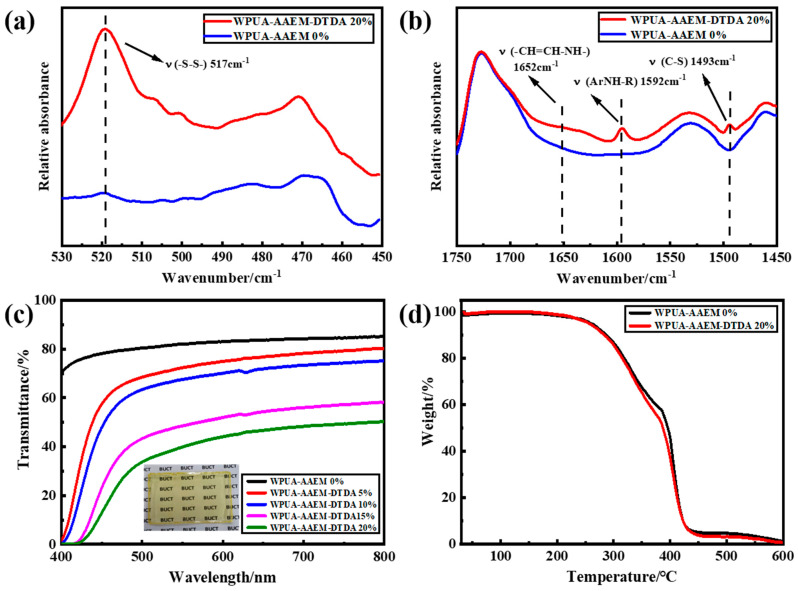
(**a**,**b**) The zoomed-in spectra of WPUA–AAEM 0% and 20% films in the wavenumber ranges of 530–450 cm^−1^ and 1750–1450 cm^−1^, respectively. (**c**) UV–vis spectra of WPUA–AAEM–DTDA *x*% films. Inset: photograph of WPUA–AAEM 10% film. (**d**) TGA profiles of WPUA–AAEM 0% and WPUA–AAEM–DTDA 20% films.

**Figure 4 molecules-28-08122-f004:**
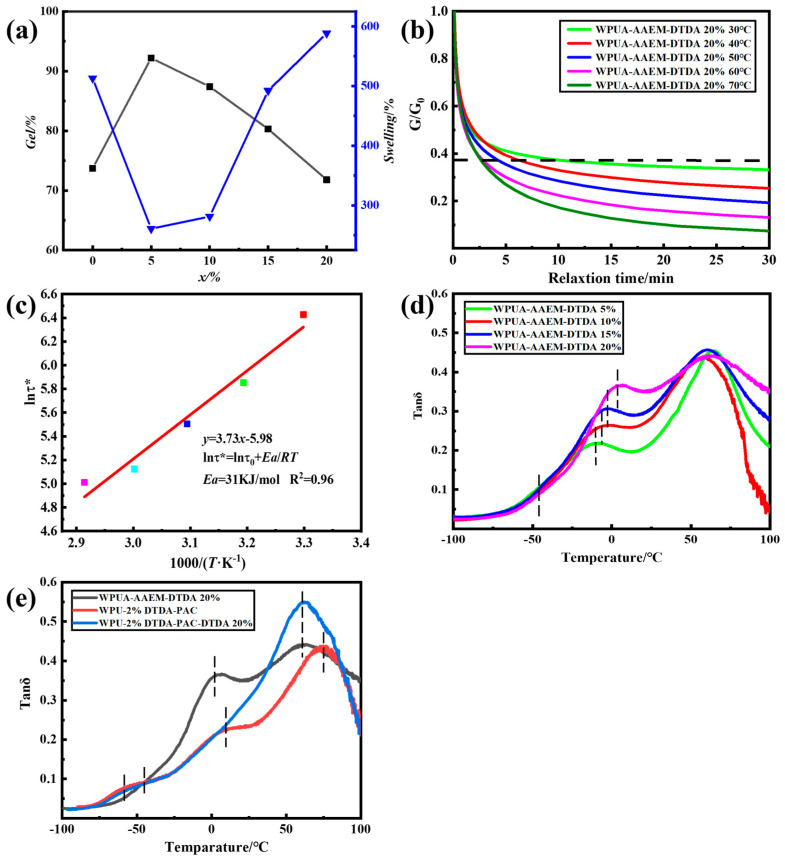
(**a**) The *gel*% and *swelling*% of WPUA–AAEM–DTDA *x*% films. (**b**) The normalized SR curves of WPUA–AAEM–DTDA 20% films at different temperatures. (**c**) Linear fitting of the relaxation time to the Arrhenius equation. (**d**) DMA tanδ–temperature spectra of WPUA–AAEM–DTDA *x*% films. (**e**) DMA tanδ–temperature spectra of the films of WPUA–AAEM–DTDA 20%, WPU–2%DTDA–PAC and WPU–2%DTDA–PAC–DTDA 20%. The black line in (**a**) represents the gel content, which is measured by the Soxhlet extraction method, and the blue represents the swelling degree, which has been written out in the article. The dashed line in (**b**) indicates the relaxation time corresponding to the relaxation process deformation to 1/e times the original. The color dots in (**c**) represent the Arrhenius equation fitting results for stress relaxation of curves at different temperatures in (**b**). The vertical dashed line in (**d**,**e**) indicates the position of the peak in the Tanδ-temperature curve, in order to see the transformation of different curves more clearly.

**Figure 5 molecules-28-08122-f005:**
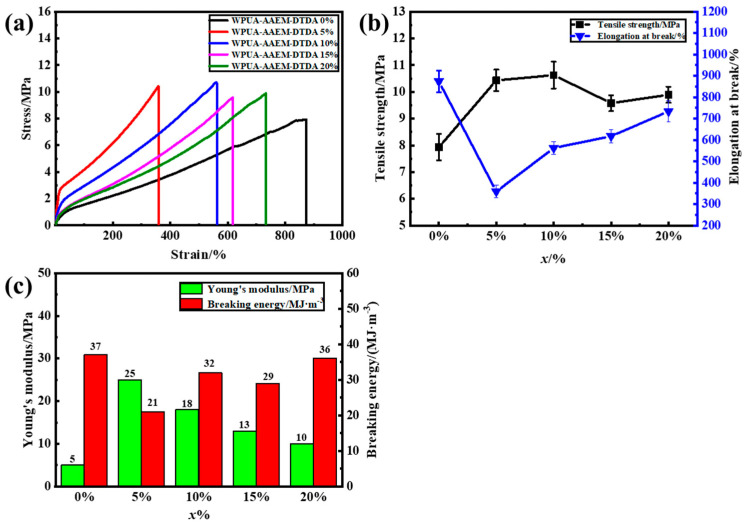
(**a**) Stress–strain curves of WPUA–AAEM–DTDA *x*% films. (**b**) Tensile strength and elongation at break of WPUA–AAEM–DTDA *x*% films. (**c**) Young’s modulus and breaking energy of WPUA–AAEM–DTDA *x*% films.

**Figure 6 molecules-28-08122-f006:**
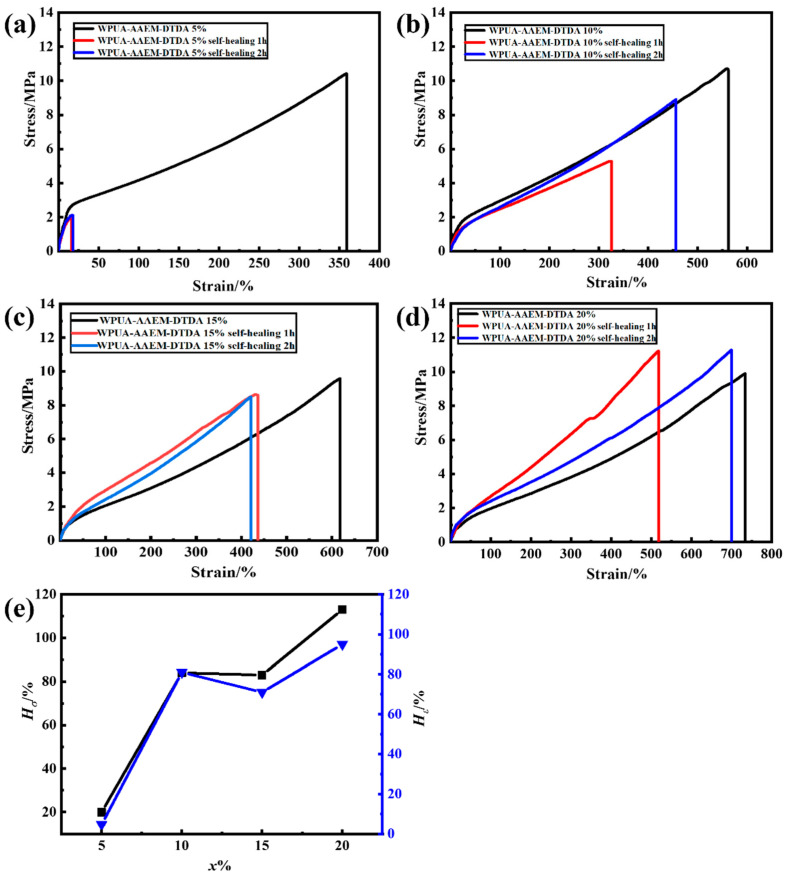
The stress–strain curves of the films self-healed at 70 °C for one or two hours in comparison with the original films of WPUA–AAEM–DTDA *x*%. (**a**) 5%, (**b**) 10%, (**c**) 15%, (**d**) 20%. (**e**) The *H_σ_* and *H_ε_* are plotted as a function of *x*%. The black and blue represent *H_σ_* and *H_ε_*, respectively. Black and blue represent self-healing efficiencies of tensile strength and elongation at break after 2h of healing at 70 °C, respectively.

**Figure 7 molecules-28-08122-f007:**
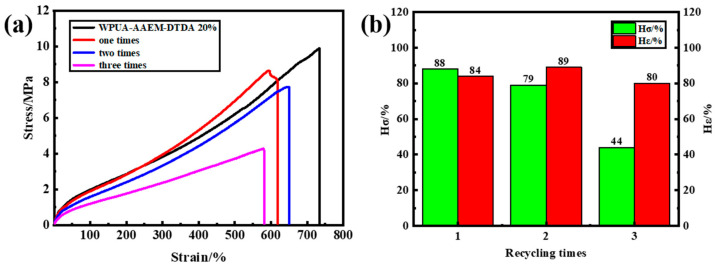
(**a**) The stress–strain curves of the WPUA–AAEM–DTDA20% film’s recovery from different reprocessing cycles. (**b**) The *H_σ_* and *H_ε_* plotted as the function of the recycling times.

**Table 1 molecules-28-08122-t001:** The properties of WPUA–AAEM *x*% composite latex.

AAEM Content (*x*)	Conv. %	Particle Size/(d·nm)	PDI	Centrifugation Stability *	Storage Stability **
0%	98%	90	0.299	Passed	≥6 months
5%	98%	105	0.194	Passed	≥6 months
10%	97%	114	0.242	Passed	≥6 months
15%	96%	138	0.368	Passed	≥6 months
20%	97%	144	0.294	Passed	≥6 months

* No stratification or precipitation was observed after 15 min of centrifugation at 3000 r/min (HB/T 4737-2014). ** The time for 100 mL of latex to keep uniform when stored in a sealed 200 mL bottle at room temperature.

## Data Availability

The data presented in this study supporting the results are available in the main text. Additional data are available upon reasonable request from the corresponding author.

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
