# Peer review of "The Post-Curing of Waterborne Polyurethane–Acrylate Composite Latex with the Dynamic Disulfide-Bearing Crosslinking Agent"

_molecules, 2023, doi:10.3390/molecules28248122_

Round 1

Reviewer 1 Report

Comments and Suggestions for Authors

Page 1, line 43:  when the complexity of the polymer aggregation structures have been involved CORRECT:  when the complexity of the polymer aggregation structures has been involved

Page 2. Line 46:  polyacrylate (PA) REMARK: PA is the IUPAC  acronym for polyamide.

Page 2, line 66: Among various DCB, the metathesis of aromatic disulfide (S-S) bonds under different kinds of stimulus have been extensively investigated. CORRECT: Among various DCBs, the metathesis of aromatic disulfide (S-S) bonds under different kinds of stimulus has been extensively investigated

Page 2, line 76: in aqueous medium, which permit the mixture CORRECT: in aqueous medium, which permits the mixture

Page 2, line 78: and especial the dynamic responses CORRECT: and especially the dynamic responses

Page 3, line 120: 1.36 of 1,4-butanediol REMARK: 1,36 g or 1.35 mL?

Page 4 line 169: with the hydrophobic parts to be 169 swelled by the inner acrylic monomers while the hydrophilic -COO- to stabilize the droplet 170 synergizing with the emulsifiers at the outermost surfaces. CORRECT: with the hydrophobic parts swelled by the inner acrylic monomers while the hydrophilic -COO - groups stabilize the droplet synergizing with the emulsifiers at the outermost surfaces.

Page 7, line 239: The gel content (gel%)  was ~73% for WPUA-AAEM 0%, which would be contributed by the intrinsic H-bonding responsible physical crosslinking. REMARK: H-bonding does not counteract the solubility of polymers in suitable solvents. Was the equilibrium state of the swelling degree achieved?

Page 7, line 250: The diamine would be excess CORRECT: The diamine would be in excess

Page 12, line 386: The DCPN characteristics were convinced by the associative type of  SR behavior in the moderate-temperature range (30°C to 70°C). In the low dosage of the  AAEM-DTDA, the role of the post-curing was mainly on the improved tensile strength and solvent resistance just like traditional curing system. BETTER: The DCPN characteristics indicated the associative type of  Swelling – Relaxation time behavior in the moderate-temperature range (30°C to 70°C). In the low dosage of the  AAEM-DTDA, the role of the post-curing was mainly improving tensile strength and solvent resistance, just like in traditional curing systems.

Page 12, line 396: could self-healed from the cut-off damage CORRECT: could self-heal from the cut-off damage

Page 12, line 397: Larger than 80% of the mechanical performance could also  maintained though two cycles of crushing-reprocessing. And ~ 600% of elongation was 398 also achieved after the third cycles. BETTER: : Larger than 80% retention of the mechanical performance could also be maintained through two cycles of crushing-reprocessing. Moreover, ~ 600% of elongation was achieved after the third cycle. 

Comments on the Quality of English Language

Included above 

Reviewer 2 Report

Comments and Suggestions for Authors

In this manuscript, PU latex with healing properties were succesfully synthesized. The work is interesting and well performed, however I believe it needs minor corrections prior to be published.

1 The English introduction should be improved, some sentences are not linear and of difficult comprehension.

2 Figure 1a monomer (the r is missing in the text.). Furthermore, is it the core shell sketch representative? Figure 2d seems to suggest a different morphology.

3 The explanation of UV-vis is not convincing. The presence of aromatic compounds does not explain the decrease of transmission, which, on the contrary, seems more related to scattering and consequently to morphology. Did the author investigated the morphology of the films (FESEM of breaking section obtained by cryo-fracture) ? This would be interesting, including a comparison with the same samples after healing treatment.

3 In DMA, E’ curves may show better the three transitions. Could the authors add this info and comment it?

Comments on the Quality of English Language

As above mentioned, please check again intro and abstract, to make those more readable.
